# Splicing Machinery Is Impaired in Oral Squamous Cell Carcinomas and Linked to Key Pathophysiological Features

**DOI:** 10.3390/ijms25136929

**Published:** 2024-06-25

**Authors:** Alba Sanjuan-Sanjuan, Emilia Alors-Perez, Marina Sanchez-Frías, José A. Monserrat-Barbudo, Mabel Falguera Uceda, Susana Heredero-Jung, Raúl M. Luque

**Affiliations:** 1Maimonides Biomedical Research Institute of Cordoba (IMIBIC), 14004 Cordoba, Spain; emi.ptt@gmail.com (E.A.-P.); marinasanchezfrias@gmail.com (M.S.-F.); joseu95mb@gmail.com (J.A.M.-B.); mfalguer@msn.com (M.F.U.); susana.herederoj@gmail.com (S.H.-J.); 2Reina Sofia University Hospital (HURS), 14004 Cordoba, Spain; 3Oral and Maxillofacial Surgery Department, HURS, 14004 Cordoba, Spain; 4Oral and Maxillofacial Surgery Department, CAMC Hospital, Charleston, WV 25301, USA; 5Department of Cell Biology, Physiology, and Immunology, University of Cordoba, 14014 Cordoba, Spain; 6CIBER Physiopathology of Obesity and Nutrition (CIBERobn), 14004 Cordoba, Spain; 7Anatomical Pathology Service, IMIBIC/HURS, 14004 Cordoba, Spain

**Keywords:** oral cancer, head and neck, therapeutic tool, biomarkers, splicing, alternative splicing, diagnostic, genetic alterations

## Abstract

Alternative splicing dysregulation is an emerging cancer hallmark, potentially serving as a source of novel diagnostic, prognostic, or therapeutic tools. Inhibitors of the activity of the splicing machinery can exert antitumoral effects in cancer cells. We aimed to characterize the splicing machinery (SM) components in oral squamous cell carcinoma (OSCC) and to evaluate the direct impact of the inhibition of SM-activity on OSCC-cells. The expression of 59 SM-components was assessed using a prospective case-control study of tumor and healthy samples from 37 OSCC patients, and the relationship with clinical and histopathological features was assessed. The direct effect of pladienolide-B (SM-inhibitor) on the proliferation rate of primary OSCC cell cultures was evaluated. A significant dysregulation in several SM components was found in OSCC vs. adjacent-healthy tissues [i.e., 12 out of 59 (20%)], and their expression was associated with clinical and histopathological features of less aggressiveness and overall survival. Pladienolide-B treatment significantly decreased OSCC-cell proliferation. Our data reveal a significantly altered expression of several SM-components and link it to pathophysiological features, reinforcing a potential clinical and pathophysiological relevance of the SM dysregulation in OSCC. The inhibition of SM-activity might be a therapeutic avenue in OSCC, offering a clinically relevant opportunity to be explored.

## 1. Introduction

Alternative splicing (AS) is a posttranscriptional process by which different exons are included in mRNA, resulting in proteomic diversity [1,2]. Systematic dysregulation of AS has recently emerged as an essential cancer hallmark, with a great potential to serve as a novel source of diagnostic, prognostic, or therapeutic tools [3,4,5,6,7,8,9,10,11,12,13,14,15,16,17,18,19,20,21]. Thus, it has been proven that a slight alteration in some of the components of the splicing machinery (spliceosome) can significantly affect the expression pattern of many essential genes and the appearance of oncogenic splicing variants [22]. The proteins that comprise the spliceosome, known as splicing factors (SFs), bind to RNA with specificity to tissue and control AS [23]. The deregulation of SFs leads to dysregulation of the splicing process and the aberrant appearance of variants that can promote cancer initiation and affect cancer cell phenotype, including proliferation, apoptosis, invasion, and metastasis of many cancer types [2,24].

The alteration in splicing machinery can be caused by mutations in some components or alterations in SF levels. These mutations generally impair the recognition of regulatory sites, thereby affecting the splicing of multiple genes, including oncogenes and tumor suppressors [2,25,26,27]. These factors can also act as survival factors that decrease drug-induced apoptosis or, on the contrary, enhance the pro-apoptotic effects of chemotherapy drugs [28].

Several studies have focused on the analysis and impact of some SFs in head and neck or oral cancer (Table 1). Their results are highly variable, even contradictory, regarding the expression pattern of some SFs in tumor tissues compared to healthy tissue. Most studies reported changes related to the upregulation or downregulation of certain spliceosomal elements, often correlating with patient survival, tumor aggressiveness parameters, and prognosis [3,10,11,12,13,14,15,20,29,30,31,32,33,34,35,36]. However, the data published so far focused on the dysregulation of the components of the splicing machinery in oral squamous cell carcinoma (OSCC) compared with healthy samples, which is quite limited, incomplete, and unclear.

In this context, antitumor drugs can target splicing machinery by addressing the spliceosome core [28,37]. Pladienolide-B (a macrocyclic lactone produced by Streptomyces sp.) and its derivatives can inhibit Splicing Factor 3B Subunit 1 (SF3B1), the most often mutated SFs across cancers and an essential spliceosome component in pre-RNA. Pladienolide-B has shown an antitumoral effect in the pancreas [37], prostate [38], pituitary [38], brain [39], and liver cancer [40]. However, the potential therapeutic impact of splicing machinery inhibition in oral cancer has not been explored.

**Table 1 ijms-25-06929-t001:** List of spliceosome components and splicing factors reported to be expressed in oral squamous cell carcinoma (OSCC) and head and neck squamous cell carcinoma (HNSCC) compared with control tissues. References from these reports are included.

Splicing Factor	Dysregulation Normal Tissue vs. Tumor	Effects on OSCC/HNSCC OS or Prognosis	References
**SRSF3**	Upregulated Downregulated	A positive relationship between SRSF3 expression and tumor grading. A significantly higher expression of the SR in patients with lymphatic metastasis Better overall survival rates.	*Peiqi* et al., 2016 [10] *Sun* et al., 2019 [11]
**SRSF5**	Upregulated	Downregulation of SRSF5 in oral squamous cell lines retarded cell growth, cell cycle progression, and tumor growth.	*Yang* et al., 2018 [12]
**SRSF9**	Upregulated Unspecified	SRSF9 overexpression seemed a hazardous factor, with no relationship with OS, DFS, clinical stage, or tumor grading. Higher expression is associated with a poor prognosis	*Liu* et al., 2022 [29] *Cao* et al., 2020 [32]
**SRSF10**	Upregulated	Overexpression of SRSF10 was closely associated with poor survival.	*Yadav* et al., 2021 [30]
**hnRNP A1**	Upregulated	hnRNPA1 is required for the growth of OSCC cells. Overexpression of hnRNP A1 may be an early pathogenic event that could be used as a new biomarker for OSCC	*Yu* et al., 2015 [13]
**hnRNP C**	Unspecified Unspecified	Higher expression was correlated with poor outcomes Higher expression is associated with a poor prognosis	*Xing* et al., 2019 [3] *Cao* et al., 2020 [32]
**hnRNP D**	Upregulated	Overexpression is associated with significantly reduced recurrence-free survival.	*Kumar* et al., 2015 [31]
**hnRNP E2**	Downregulated	low-hnRNP E2 expression level was correlated with the histological grade of differentiation.	*Roychoudhury* et al., 2007 [15]
**hnRNP H1**	Unspecified	Higher expression was correlated with poor outcomes	*Xing* et al., 2019 [3]
**hnRNP H2**	Unspecified	Higher expression was correlated with poor outcomes	*Xing* et al., 2019 [3]
**hnRNP K**	Upregulated Unspecified Upregulated Unspecified	High levels of hnRNP K were correlated with worse OS, DSS, and DFS and multiple clinicopathological factors with a poor prognosis such as advanced tumor stage, positive node stage, advanced overall stages, extracapsular spread, and large tumor depths Higher expression was correlated with poor outcomes A significant correlation between histological grades of differentiation and hnRNP K mRNA expression could not be predicted Higher expression is associated with a poor prognosis	*Matta* et al., 2009 [14] *Wu* et al., 2012 [20] *Xing* et al., 2019 [3] *Roychoudhury* et al., 2007 [15] *Cao* et al., 2020 [32]
**hnRNP L**	Upregulated	Expression promotes the proliferation, invasion, and metastasis of OSCC.	*Jia* et al., 2016 [33]
**ESRP1** **ESRP2**	Downregulated	The expression levels of both ESRP1 and ESRP2 were low in normal epithelium but upregulated in precancerous lesions and carcinoma in situ. Expression was maintained in advanced cancer cells but down-regulated on invasive fronts	*Ishii* et al., 2014 [34]
**RBM3**	Downregulated	N/A	*Martinez* et al., 2007 [35]
**NOVA1**	Downregulated Upregulated	HNSCCa HPV-negative. The lower expression was an independent poor prognosis factor for OS and PFS and related to older age, advanced pT stage, and advanced pN. HPV-positive oropharyngeal squamous cell carcinoma (SCC).	*Kim* et al., 2019 [36]
**TIA1A**	Unspecified Unspecified	Higher expression is associated with a better prognosis Higher expression was correlated with poor outcomes	*Cao* et al., 2020 [32] *Xing* et al., 2019 [3]
**TRA2B**	Upregulated	N/A	*Best* et al., 2013 [41]
**CELF2**	Unspecified	Higher expression is associated with a better prognosis	*Cao* et al., 2020 [32]

For all the reasons mentioned above, a better understanding of the regulation of splicing and OSCC tissues may help to identify novel diagnostic and prognostic biomarkers and therapeutic tools to target these tumor pathologies. However, to the best of our knowledge, no studies have reported a comprehensive analysis to ascertain whether the components of the splicing machinery are altered in OSCC. Thus, in this study, we aimed to determine—for the first time—the expression profile of a representative set of spliceosome components and SFs and their relationship with relevant clinical and histopathological parameters (stage, histological grade, tumor invasion, presence of metastasis, recurrence, overall survival, etc.) of OSCC samples and patients, as well as to assess the therapeutic potential of the inhibition of the activity of splicing machinery (using the inhibitor pladienolide-B) in primary OSCC human cell cultures.

## 2. Results

This study includes the analysis of 37 patients diagnosed with OSCC, 19 men (52%), and 18 women (48%), with a mean age of 64 ± 2 years old (range 26–86 years). Patients were followed up for at least five years. The overall survival (OS) was 60% (22/37). The disease-related death rate was 70% (11/15), with a survival rate of 45 ± 3.7 (range 2–72) months. Our cohort is comprised of 51% of patients with advanced Stage IV, 16% with Stage III, 27% with Stage II, and 6% with Stage I. 35% of our patients belonged to pT4 tumors, 24% were pT3, 35% were pT2, and 6% were pT1. The cervical lymph node involvement was positive in 43%, with pN1 in 11% and pN2 and pN3 in 16%. The recurrence analysis showed that the overall recurrence rate (RR) was 29% (10/34), the local recurrence was 23% (8/34), the regional recurrence was 23% (8/34), and both local and regional combined recurrence was 15% (5/34). The cohort’s distant metastasis rate was 12% (4/34).

### 2.1. Dysregulation of the Expression of Splicing Machinery Components in OSCC vs. Healthy Oral Cavity Samples

OSCC microfluidic array analysis of the spliceosomal landscape revealed a profound dysregulation of splicing machinery components (spliceosome and splicing factors), which significantly altered 12 of 59 components (20%) (Figure 1A). Specifically, when comparing the expression levels of the splicing machinery components analyzed between the tumor sample and control tissues (Figure 1B), we found a significant downregulation in *TRA2B*, *TIA1*, *SRSF4* (with a *p*-value < 0.05), *SRSF9*, *TRA2A* (with a *p*-value < 0.01), *ESRP1*, *NOVA1* (with a *p*-value < 0.001), and *SRSF5*, *ESRP2*, *RBM10*, and *RBM3* (with a *p*-value < 0.0001). In contrast, *SRSF10* was found to be upregulated in OSCC compared with control samples (*p* < 0.05) (Figure 1B).

Individual ROC curve analysis with the 12 spliceosome components ranged from 0.617 to 0.810 [Figure 2; i.e., SRSF10 (0.6333: *p* = 0.0628), SRSF9 (0.6533: *p* = 0.0253), SRSF5 (0.7464: *p* = 0.0004), TRA2A (0.7084: *p* = 0.0029), ESRP2 (0.8104: *p* = 0.0001), RBM10 (0.7860: *p* = 0.0001), ESRP1 (0.7793: *p* = 0.0001), RBM3 (0.7455: *p* = 0.0003), NOVA1 (0.7851: *p* = 0.0001), TRA2B (0.6173: *p* = 0.0850), TIA1 (0.6773: *p* = 0.0109), and SRSF4 (0.6416: *p* = 0.0448)]. These data demonstrate not only that the expression of the spliceosomal components *SRSF10*, *SRSF9*, *SRSF5*, *TRA2A*, *ESRP2*, *RBM10*, *ESRP1*, *RBM3*, *NOVA1*, *TRA2B*, *TIA1*, and *SRSF4* is significantly dysregulated in OSCC samples compared with control-adjacent tissues but also that they could serve as potential diagnostic biomarkers of OSCC. In this sense, further clustering and hierarchical bioinformatics analyses [i.e., Variable Importance in Projection (VIP) Score of the Partial Least Squares Discriminant Analysis (PLS-DA) analysis] in the human sample cohort analyzed revealed that the spliceosome components and splicing factors with a higher capacity of discrimination between the OSCC and control-adjacent sample groups were ESRP2, RBM10, ESRP1, RBM3, and NOVA1, being the most relevant genes for the classification model (VIP-Score > 1.8) (Figure 3A,B).

Furthermore, a multiple receiver operating characteristic (ROC) curve analysis with the expression levels of these five top discriminating components of the splicing machinery generated an area under the curve (AUC) of 0.876, 95% (CI 0.742–0.961) (Figure 3C), which demonstrated a potential capacity of discrimination of the selected components of the splicing machinery between tumor and non-tumor samples.

### 2.2. In Vivo Association between the Dysregulation of the Expression of Splicing Machinery Components in OSCC with Clinical and Pathological Data

As previously reported [42], to determine the relationship between the expression levels in OSCC tissues and the different clinical and pathological variables, we represented the expression levels of mRNA as numerical or categorical [expression level higher (>) or lower (<) median values]. It should be noted that, given the high number of analyses performed and to simplify the representation of these associations, we also decided to include only the “*p*” and corresponding “R” values of these analyses in the tables described below.

#### 2.2.1. Survival and Recurrence Data

Our analyses revealed that higher expression of TRA2A was related to better OS (*p* = 0.04; Table 2). A trend for significant association was also found for higher expressions of SRSF9 and TRA2B (*p* = 0.09, *p* = 0.09) (Table 2). Recurrence analysis showed that higher expression of SRSF9 has a lower incidence of local (*p* = 0.06) and locoregional recurrence (*p* = 0.03) (Table 2). Moreover, we found that higher expression of RBM3 had a lower incidence of distant metastasis (*p* = 0.03; Table 2).

#### 2.2.2. Staging Data

The numerical analysis showed a trend for smaller tumors (smaller pT) when expression of SRSF5, SRSF9, and TRA2A was higher [pT (*p* = 0.08); pTx2 (*p* = 0.08); Table 3]. Patients with higher expression of TRA2A, TRA2B, and TIA1 presented with less cervical nodal disease [pN−/pN+ (*p* = 0.07, 0.06, and 0.03, respectively); Table 3] and less stage [Stage x2 (*p* = 0.03, 0.09, and 0,05, respectively); Table 3]. Also, patients with higher expression of TIA1 presented with less pNx2 (*p* = 0.04; Table 3). Moreover, patients with higher expression of NOVA1 and ESRP2 presented with less and higher stages (*p* = 0.08 and 0.02, respectively; Table 3).

#### 2.2.3. Histopathological Data

Regarding histopathological factors, the higher expression of TRA2B and TIA was related to better-differentiated tumors, or G1 (*p* = 0.01 and 0,06, respectively; Table 3). TIA1 expression was statistically increased in patients with smaller depths of invasion [DOIx3 (*p* = 0.04); Table 3]. Peritumoral inflammation showed a positive correlation with the expression of SRSF5 [PTI (*p* = 0.06); PTIx2 (*p* = 0.01)], SRSF9 [PTI (*p* = 0.01); PTIx2 (*p* = 0.01)], TRA2B [PTI (*p* = 0.03); PTIx2 (*p* < 0.01)], and TIA [PTI (*p* = 0.04); PTIx2 (*p* = 0.01)]. (Table 3). Moreover, NOVA1 and ESRP2 expression were statistically increased in OSCC with an expansive front of tumor invasion compared to OSCC with an infiltrative front of tumor invasion (*p* = 0.04 and 0.06, respectively; Table 3). Similarly, our results showed that NOVA1 and ESRP2 were overexpressed in OSCC with uniform tumor invasion edges compared to poorly defined ones (*p* = 0.04 and 0.06, respectively; Table 3). Finally, we found that the expression of SRSF5, TRA2A, TRA2B, and TIA1 in OSCC was negatively correlated to the number of positive lymph nodes, the number of lymph nodes with extranodal extension (ENE+), and/or their bigger size (Table 4).

### 2.3. Antitumor Actions of an Inhibitor of the Splicing Machinery (Pladienolide-B) on Patient-Derived Primary Oral Squamous Carcinoma Cell Cultures

In the present study, and based on the previous results indicating that the expression of key spliceosomal components is consistently dysregulated in OSCC samples and that a relationship is found between some of these components and essential clinical, histopathological, and survival data, we explored whether the inhibition of the activity of the splicing machinery might influence the pathophysiology of the OSCC cells. To that end, and as previously reported in other cancer types [37,40,43,44], we performed a pharmacological experimental approach by blocking the activity of SF3B1 (a central and core component of the splicing machinery) using a specific inhibitor (pladienolide-B). First, we performed a dose-response pilot study using three different concentrations of pladienolide-B in one primary OSCC cell culture at different incubation times (Figure 4A). We found that the 100 nM dose was the most effective concentration for reducing cell proliferation rate at 24-, 48-, and 72-h of incubation (Figure 4A). Then, we used pladienolide-B (100 nM) in different OSCC cell culture specimens and demonstrated that the inhibition of the activity of the splicing machinery was able to significantly decrease the proliferation rate of OSCC cells in a time-dependent manner (Figure 4B) without significantly affecting the viability of primary cultures of normal, healthy adjacent tissues (Figure 4C).

## 3. Discussion

Oral squamous cell carcinoma, one of the most malignant tumors worldwide, continues to be a significant challenge, with many unknowns to be resolved regarding the molecular characterization of these tumors [45,46]. Thus, the high incidence, together with the hidden onset, low survival rate, and limited and inefficient treatments, clearly emphasize the necessity of identifying new molecular diagnostic, prognostic, and therapeutic tools enabling the refinement of their detection, the definition tumor behavior, and the development of new treatments for this cancer type. In this context, splicing dysregulation is a hallmark of many cancer types [47]. It has emerged as a novel source for identifying new biomarkers for the diagnosis and prognosis of numerous cancers, including OSCC 3, 10–15, 20, 28–35. However, to our knowledge, these studies in OSCC have not comprehensively explored the global dysregulations of spliceosomal components and splicing factors in OSCC. A leading cause for this leading role of the splicing process in cancer resides in mutations and altered expression in splicing machinery components, which can modify the splicing patterns of multiple genes [48]. Therefore, in this study, we aimed to investigate the status of the splicing machinery in OSCC vs. non-tumor adjacent tissue, which is linked to clinical and/or pathological features and might exert functionally relevant roles in OSCC to identify novel prognostic biomarkers and therapeutic targets in this disease.

Our results demonstrate a drastic dysregulation of the expression profile of the components belonging to the splicing machinery in a well-characterized cohort of OSCC compared with control-adjacent tissues, where a representative set of these components was significantly altered [12 out of 59 components (20%)]. Specifically, we found a downregulation of SRSF4, SRSF5, SRSF9, NOVA1, ESRP1, ESRP2, RBM3, RBM10, TRA2A, TRA2B, and TIA1, and an upregulation of SRSF10 expression levels. These differences observed in the expression profile of the splicing machinery in OSCC tissue and its surrounding normal tissue were expected, in line with those observed by our group in different cancer types [39,44,49,50]. In this context, the expression levels of specific splicing factors in OSCC samples vary in the literature (Table 1). Some splicing factors have been described as upregulated, downregulated, or even oppositely altered [3,10,32,51]. Our study found an overall downregulation of most of the splicing factors in OSCC samples, inviting us to explore these molecules further as potential diagnostic and prognostic biomarkers (see below for further discussion). These results were consistent with prior studies that also found down-regulation of NOVA1 [36], RBM3 [35], ESRP1, and ESRP2 [34] in OSCC, but, to the best of our knowledge, this is the first study to describe the downregulation of SRSF4, SRSF9, RBM10, TRA2A, TRA2B, and TIA1 in OSCC. On the other hand, we found SRSF10 expression to be upregulated in our cohort of OSCC samples, which was also consistent with a previous study reported with head and neck samples [30]. However, in our study, SRSF10 upregulation (whose change was visually inapparent as can be observed in Figure 1B) did not show any correlation with OSCC survival or histopathological risk factors, while SRSF10 was reported to play a crucial role in head and neck tumorigenesis in the previous study [30]. This difference may be due to the intrinsic phenotypic differences between samples as well as protocol and race differences.

Notably, ROC curve analysis revealed that the majority of these components (SRSF4, SRSF5, SRSF9, NOVA1, ESRP1, ESRP2, RBM3, RBM10, TRA2A, and TIA1) could serve as potential diagnostic biomarkers of OSCC [AUC obtained ranged from 0.642 (for SRSF4) to 0.810 (for ESRP2)]. Moreover, the VIP score analysis revealed that the spliceosome components with a higher discrimination capacity between OSCC and healthy samples were ESRP1, ESRP2, RBM3, RBM10, and NOVA1. This invites us to explore these molecules further as potential diagnostic biomarkers. In support of this idea, we found that the potential diagnostic ability clearly improved when the ROC curve analysis was performed with these top five spliceosome components (i.e., with a higher capacity of discrimination: ESRP1, ESRP2, RBM3, RBM10, and NOVA1), obtaining an AUC of 0.88.

Therefore, the next logical step was to find correlations between the relevant spliceosome components in this study and clinical or pathological parameters, since this could also guide the identification of relevant prognostic biomarkers. In fact, the potential utility of some of the altered spliceosome components in OSCC as prognostic biomarkers is further supported by their direct association between their levels and relevant clinical or pathological features of aggressiveness. Specifically, we found that overall survival was positively correlated with higher expression of *TRA2A*, *TRA2B*, and *SRSF9*. Interestingly, these splicing factors were downregulated in tumor samples compared with healthy adjacent tissues, and their expression in the OSCC tissue was associated with better OS. This is the first study demonstrating the relationship between these splicing factors, oral cancer, and better survival. Notably, the levels of TRA2B, SRSF9, and RBM3 were also associated with less recurrence or distant metastasis, suggesting that these splicing factors might have pathophysiological relevance in this tumor pathology and suggesting a causal link between dysregulation of these splicing factors and OSCC aggressiveness.

In addition, the expression of SRSF9, TRA2A, and TRA2B was also associated with improved OS, less recurrence or distant metastasis, and other splicing factors such as SRSF5 and TIA1 were related to clinical and histopathological features of a better outcome, including fewer cervical nodal disease (pN), less ENE+ lymph nodes, smaller tumors (pT), a lower grade of differentiation, a lower DOI, or a higher PTI. Furthermore, TIA expression levels were also associated with other key histopathological factors related to better outcomes, such as a lower grade of differentiation, a higher PTI, or a lower DOI. These results are in accordance with a previous study indicating that higher expression of TIA was associated with a better prognosis [32]. Although TRA2B expression has been previously described as altered in head and neck cancers, no information related to its impact on OSCC has been previously reported. Consequently, this is the first study to describe a more detailed knowledge of the histopathological relationship between TRA2B, TIA, and OSCC patients, as well as the first one for TRA2A and OSCC.

Splicing factors are considered molecular tools for the chemotherapy response, acting as either prosurvival factors that diminish drug-induced apoptosis or, oppositely, potentiate the pro-apoptotic effects of chemotherapeutics [52]. The specific influence of individual splicing factors on the efficacy of chemotherapy drugs used in head and neck cancer has only been studied in the case of SRSF3, which was shown to be associated with reduced sensitivity of cancer cells to Paclitaxel (PTX) treatment [11]. Other splicing factors have also been associated with PTX efficacy, such as TRA2A promoting resistance to PTX in breast cancer [11]. In this context, our results describe for the first time the association between TRA2A and OSCC oral squamous cell carcinoma, among other splicing factors, and their relationship might unveil the role of these newly described splicing factors as therapeutic targets in OSCC. In line with this, several reports have indicated that cancer cells are particularly vulnerable to splicing alterations. These changes might be relevant from a therapeutic point of view since the transcriptomic landscape of cancer cells makes them particularly vulnerable to the pharmacological inhibition of splicing [28,53]. In support of this idea, our study also provides an initial, unprecedented proof-of-concept on the suitability of splicing dysregulation as a novel potential target for OSCC treatment by demonstrating that the pharmacological impact of inhibiting the splicing process has significant beneficial consequences in OSCC cells. Specifically, we tested pladienolide-B’s direct in vitro effect in primary OSCC cell cultures. We demonstrated, for the first time, that inhibition of the splicing machinery activity significantly inhibited cell proliferation in OSCC but not healthy adjacent cells, which compares well with recent data from our group showing that pladienolide-B reduced proliferation rates in the prostate, pituitary, liver, pancreas, and brain tumors [37,38,39,40].

The present study has some limitations: (i) the limited number of cases analyzed that we would like to continue increasing for future investigations and following these patients for a proper analysis of the impact of spliceosome components on patient’s survival and other relevant clinical/pathological characteristic as well as multivariable analysis; and, (ii) due to the limitation in the number of tumor tissues that can be colleted, and in the number of primary cells that can be obtained from the tumoral and healthy-adjacent tissues obtained, we could not perfom westen-blot analyses or include studies aimed to unravel the molecular/functional consequences and signaling pathways underlying the link between the dysregulation of these splicing factors and clinical or histopathological features in OSCC patients. Nevertheless, it is clear that to solve these limitations and further support our findings, we plan to analyze a larger tumor cohort in more detail, and studies are already ongoing aimed at that goal. This is important because it is well-recognized that the splicing process and its regulation are highly relevant for understanding every hallmark of cancer, to the point that splicing alterations constitute another cancer hallmark [54,55,56].

## 4. Conclusions

Our results unveiled new conceptual and functional avenues in OSCC, with potential therapeutic implications, by demonstrating for the first time a dysregulation of the splicing machinery in OSCC compared with healthy-adjacent oral cavity tissues. This is likely relevant clinically because the dysregulation is directly associated with key pathophysiological features of OSCC. Moreover, our data highlight the inhibition of the splicing machinery as a putative and efficient pharmacological target in OSCC, offering a clinically relevant opportunity worth exploring in humans. Therefore, these findings underscore the potential of the splicing machinery and the splicing process as a novel source to better understand OSCC biology and identify candidate biomarkers and actionable targets.

## 5. Materials and Methods

### 5.1. Patient Data and Samples Collection

The Ethics Committee approved the study, and written informed consent was obtained from all the patients (see Institutional Review Board Statement and Informed Consent Statement at the end of the manuscript). A prospective observational case-control study was performed with 37 patients diagnosed with OSCC, 19 men (52%), and 18 women (48%), with a mean age of 64 ± 2-years-old (range 26–86 years). SCC originated from the tongue in 20 out of 37 patients (54%) and from the floor of the mouth in 6 patients (16%). In the rest, it was found in the alveolar ridge or hard palate in 5 patients (14%), in the buccal mucosa in 3 patients (8%), in the retromolar trigone in 2 patients (5%), and in 1 patient (3%) the origin was the lower lip. Clinical variables were obtained from the clinical chart. Specifically, stage, histological grade, tumor pT stage, cervical metastasis or pN, depth of invasion (DOI), perineural (PNI) or lymphovascular invasion (LVI), peritumoral inflammation (PTI) (absent, mild, moderate, severe), pattern of tumor invasion (infiltrative, exophytic), lymph node size, and extranodular extension (ENE+) were recorded. For better analysis, variables such as stage, DOI, pT, pN, and PTI were divided into subcategories or dichotomous categories. Disease-overall survival (OS) and Disease-free survival (DFS) were calculated. Three patients who died before six months due to perioperative complications were classified as “lost data” for recurrence analysis. Overall recurrence rate (RR), local recurrence, regional recurrence, local and regional combined, and distant metastasis were calculated.

OSCC tumor tissue samples (case) were obtained from the surgical specimen after resection. Healthy adjacent tissue samples (control) were obtained within the same patient from the buccal mucosa with a distance from the tumor greater than 2 cm. Then, both specimens were immediately deposited in a cold culture medium and transported to the laboratory. The control sample and a fragment of the tumor tissue were frozen at −80 °C for subsequent RNA isolation, retrotranscription, and expression analysis by quantitative PCR (qPCR) based on microfluidic technology (see below). The remaining tumor tissue was used to perform cell cultures (see below). The tissue sample was consistently obtained safely and ethically, and it did not interfere with the pathologist’s work.

### 5.2. RNA Isolation and Retrotranscription (RT)

Total RNA from all samples was extracted simultaneously using the RNase-Free DNase Set (Qiagen, Limburg, The Netherlands), according to manufacturer instructions, as previously reported [39,42,57]. The amount of RNA recovered and its purity were determined using the Nanodrop One Spectrophotometer (Thermo Fisher Scientific, Madrid, Spain). One μg of total RNA was retrotranscribed to cDNA with the First-Strand Synthesis kit (MRI Fermentas, Hanover, MD, USA) using random hexamer primers in a 20 μL volume, as previously reported [58].

### 5.3. Analysis of Splicing Machinery Components by qPCR Dynamic Array

A qPCR Dynamic Array (Fluidigm, South San Francisco, CA, USA) based on microfluidic technology was employed to simultaneously measure the expression levels of 59 genes (including 3 housekeeping genes; see below) in 37 OSCC samples and normal healthy-adjacent tissues. Specifically, this custom array included components of the major spliceosome (*n* = 10) and minor (*n* = 4) spliceosome, associated SFs (*n* = 42), and three housekeeping genes (*ACTB*, *GAPDH*, and *HPRT*). We performed a preamplification, an exonuclease treatment, and the qPCR dynamic array following the manufacturer’s instructions as previously described [59], using the Biomark system (Fludgim). The data were processed with Real-Time PCR Analysis Software 3.0 (Fluidigm). To control for variations in the efficiency of the retrotranscription reaction, mRNA copy numbers of the different transcripts analyzed were adjusted by a normalization factor (NF), calculated with the expression levels of 3 housekeeping genes [actin-beta, (ACTB), hypoxanthine-guanine phosphoribosyltransferase 1 (HPRT), and glyceraldehyde 3-phosphate (GAPDH)], using the Genorm 3.3 software, as previously reported [42]. This selection was based on the stability of these housekeeping genes among the experimental groups to be compared, wherein the expression of these housekeeping genes was not significantly different among groups.

### 5.4. Primary OSCC Cell Culture

As mentioned before, and when possible, a piece of the OSCC tissues and its normal healthy adjacent tissue were collected after surgery in sterile, cold PBS 1x (Omega Scientific, Tarzana EEUU, CA, USA) with a 1% antibiotic-antimycotic solution and immediately dispersed into single cells under sterile conditions by a mechanic/enzymatic protocol as previously reported [42]. The single cells were seeded in RPMI 1640 (ThermoFisher Scientific) medium supplemented with 10% fetal bovine serum (FBS), 1% antibiotic-antimycotic, and 2mML-glutamine in plates previously coated with poly-L-lysine to enhance cell adherence. Cell number and viability (always higher than 95%) were determined by the trypan blue dye exclusion method (American Type Culture Collection, Manassas, VA, USA) in a Neubauer Chamber.

### 5.5. In Vitro Cell Proliferation Assay

Cell proliferation in response to the treatment of Pladienolide-B (Santa Cruz, Heidelberg, Germany) was measured using Alamar-blue reagent after seeding 10,000 cells per well in a 96-well plate. Briefly, cells were used in a serum-free medium to achieve cell synchronization. Then, cell proliferation was measured at 0, 24, 48, and 72 h using the FlexStation3 system (Molecular Devices, Sunnyvale, CA, USA). A proliferation assay was used before performing a Pladienolide-B dose response (0.01 nM, 1 nM, and 100 nM); and the dose selected was 100 nM. All assays were repeated a minimum of three times on independent days.

### 5.6. Statistical and Bioinformatical Analysis

All data are expressed as the mean ± SEM. Statistical analysis was done using SPSS (IBM, New York, NY, USA) and GraphPad Prism (La Jolla, CA, USA). Normality was assessed using the Shapiro or Kolmogorov-Smirnov tests and by visual inspection of the shapes of histograms. We evaluate the heterogeneity of variance using the Kolmogorov-Smirnov test to compare the difference between the means of the gene’s expression levels in tumor tissue and healthy tissues within the same patient. Consequently, parametric (Student-t) or nonparametric (Mann-Whitney U) tests were implemented. A one-way ANOVA analysis was performed to explore statistical differences between the two groups.

Statistical analysis of ROC curves was performed by calculating each element’s area under the curve (AUC) and comparing it with the AUC of the reference line using the Student’s *t*-test. Heatmaps, VIP score, and PLS-DA analysis were performed using MetaboAnalyst 3.0. The statistical studies from functional assays were assessed using a paired parametric t-test or one-way ANOVA test, followed by Dunnett’s test for multiple comparisons. Data were expressed as mean ± SEM. Clinical correlations were evaluated by the unpaired nonparametric Mann-Whitney test or the Spearman test.

Survival curves were calculated using Kaplan-Meier analysis, and the log-rank test was used to compare OS and recurrence according to different variables. Parametric or nonparametric tests were used to analyze the relationship between clinical and staging data, histopathological analysis, and expression levels of splicing factors. Pearson or Spearman correlation analyses were used to assess the relationship between numerical variables. *p*-values ≤ 0.05 were considered statistically significant. A significant trend was indicated when *p*-values ranged between >0.05 and <0.1.

## Figures and Tables

**Figure 1 ijms-25-06929-f001:**
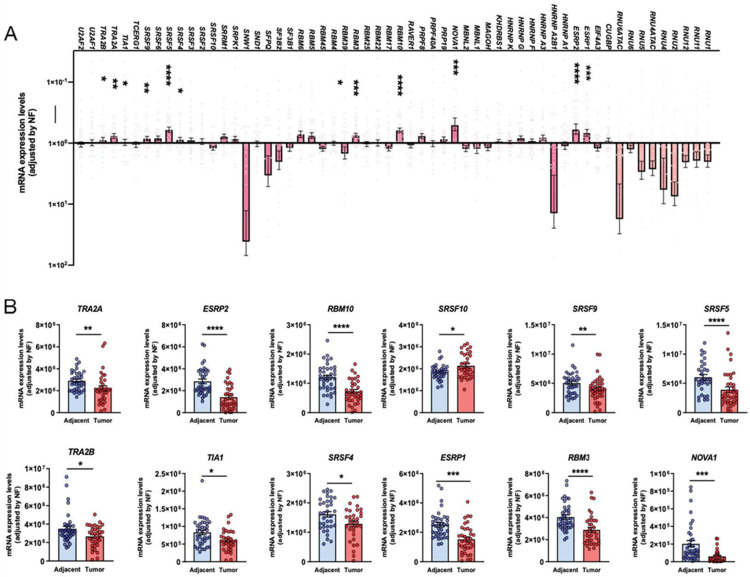
(**A**) mRNA expression levels of all the components of the splicing machinery analyzed in OSCC samples compared with non-tumoral adjacent tissue. (**B**) Individual description of mRNA expression levels of the statistically significant dysregulated spliceosome components in OSCC compared with non-tumoral adjacent tissue. Data are represented as mRNA levels normalized by a normalization factor (NF), calculated with the expression levels of three housekeeping genes: [actin-beta, (ACTB), hypoxanthine-guanine phosphoribosyltransferase 1 (HPRT), and glyceraldehyde 3-phosphate (GAPDH). Asterisks indicate significant differences (*, *p* < 0.05; **, *p* < 0.01; ***, *p* < 0.001; ****, *p* < 0.0001).

**Figure 2 ijms-25-06929-f002:**
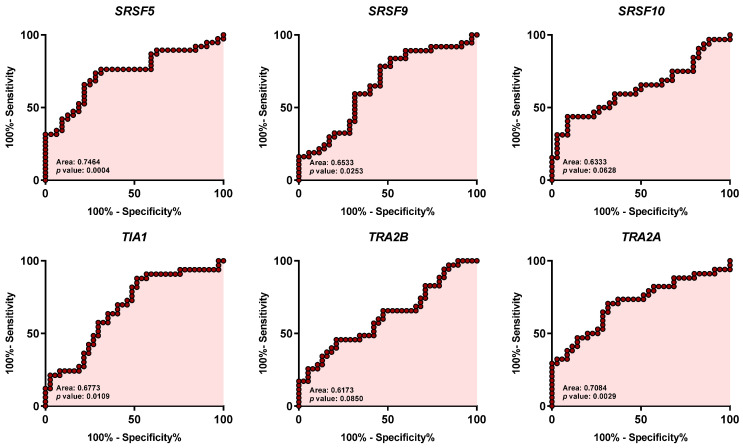
Receiver Operating Characteristic (ROC) curve analysis of significantly dysregulated splicing machinery components in OSCC samples compared with non-tumoral adjacent tissue. Specific AUC (Area Under the ROC Curve) obtained ranged from 0.617 to 0.810 [i.e., SRSF10 (0.6333: *p* = 0.0628), SRSF9 (0.6533: *p* = 0.0253), SRSF5 (0.7464: *p* = 0.0004), TRA2A (0.7084: *p* = 0.0029), ESRP2 (0.8104: *p* < 0.0001), RBM10 (0.7860: *p* < 0.0001), ESRP1 (0.7793: *p* < 0.0001), RBM3 (0.7455: *p* = 0.0003), NOVA1 (0.7851: *p* < 0.0001), TRA2B (0.6173: *p* = 0.0850), TIA1 (0.6773: *p* = 0.0109), and SRSF4 (0.6416: *p* = 0.0448)].

**Figure 3 ijms-25-06929-f003:**
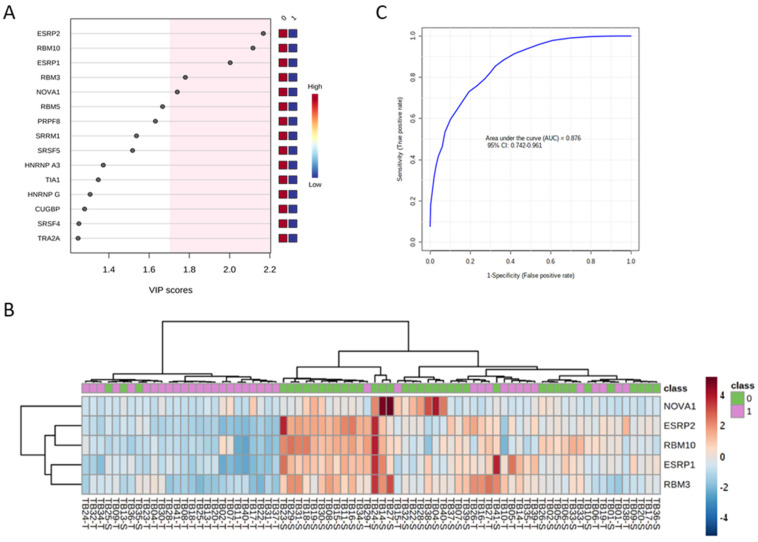
Discriminatory value of the top 5 genes of splicing machinery components and splicing factors in OSCC (ESRP1, RBM10, ESRP2, RBM3, and NOVA1). (**A**) VIP scores obtained from Partial Least Squares Discriminant Analysis (PLS-DA) of the spliceosome components analyzed in OSCC vs. non-tumoral adjacent tissues. (**B**) Unsupervised clustering analysis of mRNA expression levels of the 5 top discriminating spliceosome components and splicing factors (ESRP1, RBM10, ESRP2, RBM3, and NOVA1; shown as a hierarchical heatmap) in OSCC samples (1: shown in green) compared with non-tumoral adjacent tissue (0: shown in red). (**C**) Receiver operating characteristic (ROC) curve analysis with the expression levels of the 5 top discriminating spliceosome components and splicing factors (ESRP1, RBM10, ESRP2, RBM3, and NOVA1).

**Figure 4 ijms-25-06929-f004:**
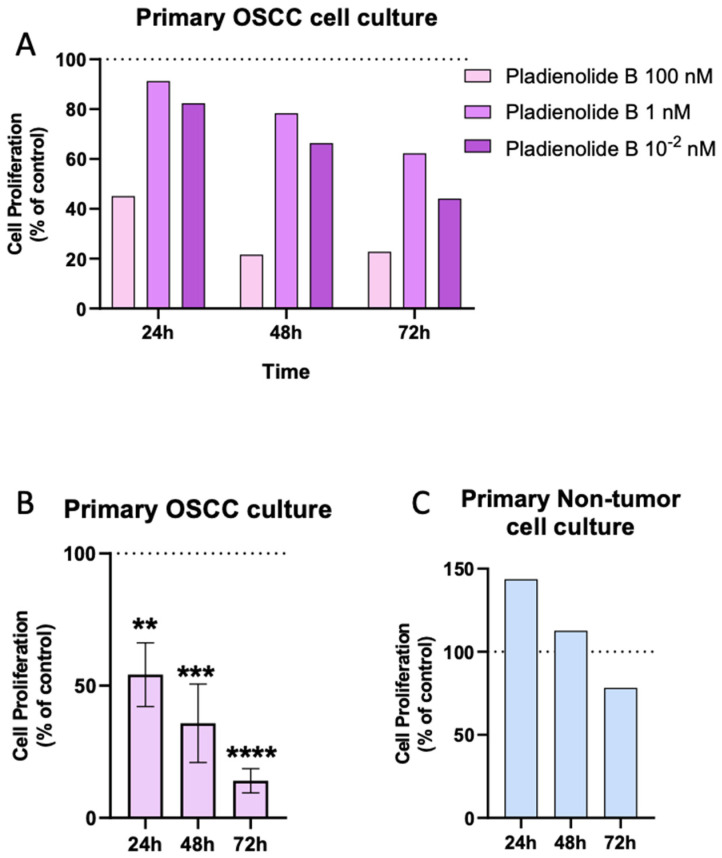
Pharmacological inhibition of the splicing machinery with Pladienolide B in primary cell cultures of OSCC and normal healthy adjacent tissues. (**A**) Proliferation rate in response to different doses of Pladienolide B (0.01, 1, and 100 nM) in primary OSCC cell cultures compared to vehicle-treated control cells (control set at 100%; *n* = 1). (**B**) Proliferation rate in response to Pladienolide B administration in primary OSCC cell cultures (*n* = 3) and (**C**) in primary non-tumor cell cultures (*n* = 3). The control set as is 100%, represented as a dotted line. Asterisks indicate significant differences (** *p* < 0.01; *** *p* < 0.001; **** *p* < 0.0001).

**Table 2 ijms-25-06929-t002:** In vivo association between the expression of spliceosome components in OSCC and Overall Survival (OS), Recurrence Rate (RR), and Distant Metastasis. Spliceosome component expression is expressed as categorical with a “>/< median” analysis. *p*-values are calculated with a log-rank test for the analysis between >/< median analysis and OS, overall RR, Local RR, Regional RR, Local and Regional RR, and Distant Metastasis. (−) negative correlation; (+), positive correlation.

	*OS*	*RR*	*Local RR*	*Regional RR*	*Local and* *Regional RR*	*Distant Metastasis*	*Test*
** *SRSF4* ** ** *>/< median* **	*p* = 0.67 R −0.08	*p* = 0.35 R 0.21	*p* = 0.93 R 0.01	*p* = 0.81 R 0.07	*p* = 0.33 R −0.19	*p* = 0.76 R 0.11	Log-rank
** *SRSF5* ** ** *>/< median* **	*p* = 0.21 R 0.18	*p* = 0.81 R 0.09	*p* = 0.98 R 0.04	*p* = 0.76 R −0.03	*p* = 0.46 R −0.10	*p* = 0.40 R 0.16	Log-rank
** *SRSF9* ** ** *>/< median* **	***p* = 0.09** R 0.12	*p* = 0.27 R −0.22	***p* = 0.06 (−)** **R −0.32**	*p* = 0.27 R −0.23	***p* = 0.03 (−)** **R −0.38**	*p* = 0.38 R 0.09	Log-rank
** *SRSF10* ** ** *>/< median* **	*p* = 0.27 R −0.1	*p* = 0.75 R 0.06	*p* = 0.93 R 0.02	*p* = 0.72 R −0.07	*p* = 0.36 R −0.16	*p* = 0.43 R −0.16	Log-rank
** *NOVA1* ** ** *>/< median* **	*p* = 0,25 R 0.03	*p* = 0.72 R 0.01	*p* = 0.58 R −0.07	*p* = 0.28 R −0.14	*p* = 0.13 R −0.25	*p* = 0.19 R −0.18	Log-rank
** *RBM3* ** ** *>/< median* **	*p* = 0.14 R 0.19	*p* = 0.54 R −0.09	*p* = 0.58 R 0.14	*p* = 0.17 R −0.24	*p* = 0.81 R −0.01	***p* = 0.03 (−)** **R −0.40**	Log-rank
** *RBM10* ** ** *>/< median* **	*p* = 0.64 R 0.05	*p* = 0.85 R 0.07	*p* = 0.50 R −0.12	*p* = 0.72 R 0.09	*p* = 0.54 R −0.12	*p* = 0.52 R 0.15	Log-rank
** *ESRP1* ** ** *>/< median* **	*p* = 0.31 R 0.14	*p* = 0.73 R −0.1	*p* = 0.86 R −0.03	*p* = 0.75 R −0.10	*p* = 0.91 R −0.02	*p* = 0.40 R 0.08	Log-rank
** *ESRP2* ** ** *>/< median* **	*p* = 0.23 R 0.28	*p* = 0.52 R −0.12	*p* = 0.79 R −0.04	*p* = 0.53 R −0.12	*p* = 0.83 R −0.03	*p* = 0.63 R 0.07	Log-rank
** *TRA2A* ** ** *>/< median* **	***p* = 0.04 (+)** R 0.32	*p* = 0.32 R −0.16	*p* = 0.20 R −0.23	*p* = 0.32 R −0.15	*p* = 0.17 R −0.25	*p* = 0.89 R 0.06	Log-rank
** *TRA2B* ** ** *>/< median* **	***p* = 0.09** R 0.01	*p* = 0.14 R −0.19	*p* = 0.27 R −0.14	*p* = 0.12 R −0.21	*p* = 0.24 R −0.18	*p* = 0.36 R −0.09	Log-rank
** *TIA1* ** ** *>/< median* **	*p* = 0.75 R −0.2	*p* = 0.09 R 0.36	*p* = 0.10 R 0.33	*p* = 0.28 R 0.23	*p* = 0.35 R 0.19	*p* = 0.59 R 0.11	Log-rank

**Table 3 ijms-25-06929-t003:** In vivo relationship between the expression of spliceosome components in OSCC, staging, and histopathological data. Nonparametric Kruskal-Wallis and U-Mann Whitney tests analyzed the relationship between splicing numerical expression and staging data.

	*pT*	*pTx2*	*pN*	*pNx4*	*pN x2*	*pN-/pN+*	*Stage*	*Stagex2*
** *SRSF4* ** ** *Numerical* **	*p* = 0.41 R 0.04	*p* = 0.37 R 0.15	*p* = 0.12 R −0.19	*p* = 0.09 R −0.18	*p* = 0.98 R −0.01	*p* = 0.29 R −0.18	*p* = 0.64 R 0.07	*p* = 0.87 R 0.27
** *SRSF5* ** ** *Numerical* **	*p* = 0.13 R −0.37	***p* = 0.08 (−)** R −0.29	*p* = 0.26 R −0.18	*p* = 0.38 R −0.17	*p* = 0.57 R −0.09	*p* = 0.15 R −0.23	*p* = 0.58 R −0.27	*p* = 0.11 R −0.26
** *SRSF9* ** ** *Numerical* **	***p* = 0.08 (−)** R −0.36	*p* = 0.15 R −0.24	*p* = 0.70 R −0.08	*p* = 0.87 R −0.07	*p* = 0.95 R −0.01	*p* = 0.59 R −0.08	*p* = 0.47 R −0.17	*p* = 0.30 R −0.16
** *SRSF10* ** ** *Numerical* **	*p* = 0.23 R 0.09	*p* = 0.49 R 0.12	*p* = 0.11 R −0.02	*p* = 0.33 R −0.01	*p* = 0.44 R 0.14	*p* = 0.80 R −0.04	*p* = 0.13 R 0.16	*p* = 0.80 R 0.04
** *NOVA1* ** ** *Numerical* **	*p* = 0.27 R −0.24	*p* = 0.12 R −0.26	*p* = 0.77 R −0.18	*p* = 0.60 R −0.17	*p* = 0.22 R −0.20	*p* = 0.48 R −0.12	***p* = 0.08 (−)** R −0.25	*p* = 0.18 R −0.22
** *RBM3* ** ** *Numerical* **	*p* = 0.88 R −0.07	*p* = 0.90 R −0.02	*p* = 0.13 R −0.16	*p* = 0.12 R −0.16	*p* = 0.88 R 0.02	*p* = 0.18 R −0.23	*p* = 0.42 R −0.05	*p* = 0.51 R −0.11
** *RBM10* ** ** *Numerical* **	*p* = 0.98 R −0.02	*p* = 0.77 R −0.04	*p* = 0.76 R −0.12	*p* = 0.91 R −0.11	*p* = 0.51 R −0.11	*p* = 0.50 R −0.11	*p* = 0.14 R −0.01	*p* = 0.54 R −0.10
** *ESRP1* ** ** *Numerical* **	*p* = 0.12 R −0.01	*p* = 0.57 R 0.09	*p* = 0.12 R −0.01	*p* = 0.40 R 0.03	*p* = 0.35 R 0.15	*p* = 0.89 R −0.02	***p* = 0.02 (+)** R 0.11	*p* = 0.46 R 0.12
** *ESRP2* ** ** *Numerical* **	*p* = 0.19 R −0.07	*p* = 0.61 R 0.08	*p* = 0.16 R −0.07	*p* = 0.51 R −0.07	*p* = 0.82 R 0.03	*p* = 0.45 R −0.12	*p* = 0.08 R 0.06	*p* = 0.42 R 0.13
** *TRA2A* ** ** *Numerical* **	***p* = 0.08 (−)** R −0.40	*p* = 0.13 R −0.26	*p* = 0.31 R −0.28	*p* = 0.29 R −0.28	*p* = 0.37 R −0.15	***p* = 0.07 (−)** R −0.31	*p* = 0.40 R −0.35	***p* = 0.03 (−)** R −0.38
** *TRA2B* ** ** *Numerical* **	*p* = 0.40 R −0.25	*p* = 0.16 R −0.24	*p* = 0.28 R −0.34	*p* = 0.19 R −0.33	*p* = 0.17 R −0.23	***p* = 0.06 (−)** R −0.33	*p* = 0.71 R −0.26	***p* = 0.09 (−)** R −0.28
** *TIA1* ** ** *Numerical* **	*p* = 0.36 R −0.19	*p* = 0.15 R −0.25	*p* = 0.32 R -−0.41	*p* = 0.15 R −0.40	***p* = 0.04 (−)** R −0.36	***p* = 0.03 (−)** R −0.37	*p* = 0.41 R −0.30	***p* = 0.05 (−)** R −0.33
	** *G* **	** *DOIx3* **	** *PTI* **	** *PTIx2* **	** *PNI* **	** *LVI* **	** *Invasion Front* **	** *Uniformity* **
** *SRSF4* ** ** *Numerical* **	*p* = 0.21 R −0.21	*p* = 0.80 R 0.06	*p* = 0.27 R 0.23	*p* = 0.10 R 0.28	*p* = 0.84 R -0.03	*p* = 0.91 R −0.01	*p* = 0.48 R 0.12	*p* = 0.48 R 0.12
** *SRSF5* ** ** *Numerical* **	*p* = 1 R 0.00	*p* = 0.24 R −0.27	***p* = 0.06 (+)** R 0.39	***p* = 0.01 (+)** R 0.41	*p* = 0.23 R −0.15	*p* = 0.56 R −0.09	*p* = 0.81 R 0.03	*p* = 0.81 R 0.03
** *SRSF9* ** ** *Numerical* **	*p* = 0.96 R −0.01	*p* = 0.21 R −0.26	***p* = 0.01 (+)** R 0.28	***p* = 0.01 (+)** R 0.40	*p* = 0.36 R −0.19	*p* = 0.40 R −0.14	*p* = 0.30 R 0.17	*p* = 0.30 R 0.17
** *SRSF10* ** ** *Numerical* **	*p* = 0.84 R −0.03	*p* = 0.11 R 0.16	*p* = 0.61 R 0.12	*p* = 0.42 R 0.14	*p* = 0.12 R 0.27	*p* = 0.20 R 0.23	*p* = 0.36 R 0.16	*p* = 0.36 R 0.16
** *NOVA1* ** ** *Numerical* **	*p* = 0.24 R −0.19	*p* = 0.39 R −0.14	*p* = 0.11 R 0.21	*p* = 0.16 R 0.24	*p* = 0.88 R 0.02	*p* = 0.76 R −0.05	***p* = 0.04 (+)** R 0.34	***p* = 0.04 (+)** R 0.34
** *RBM3* ** ** *Numerical* **	*p* = 0.83 R −0.03	*p* = 0.75 R −0.12	*p* = 0.46 R 0.11	*p* = 0.46 R 0.12	*p* = 0.59 R −0.09	*p* = 0.12 R −0.26	*p* = 0.13 R 0.26	*p* = 0.13 R 0.26
** *RBM10* ** ** *Numerical* **	*p* = 0.17 R −0.23	*p* = 0.59 R −0.15	*p* = 0.31 R −0.02	*p* = 0.64 R 0.07	*p* = 0.27 R −0.18	*p* = 0.26 R −0.18	*p* = 0.65 R 0.07	*p* = 0.65 R 0.07
** *ESRP1* ** ** *Numerical* **	*p* = 0.69 R −0.06	*p* = 0.82 R −0.04	*p* = 0.56 R 0.04	*p* = 0.73 R 0.05	*p* = 0.69 R 0.06	*p* = 0.57 R −0.09	*p* = 0.12 R 0.26	*p* = 0.12 R 0.26
** *ESRP2* ** ** *Numerical* **	*p* = 0.78 R −0.04	*p* = 0.68 R −0.14	*p* = 0.47 R 0.13	*p* = 0.31 R 0.17	*p* = 0.66 R −0.07	*p* = 0.43 R −0.13	***p* = 0.06 (+)** R 0.30	***p* = 0.06 (+)** R 0.30
** *TRA2A* ** ** *Numerical* **	*p* = 0.80 R 0.04	*p* = 0.22 R −0.29	*p* = 0.17 R 0.24	*p* = 0.06 R 0.32	*p* = 0.87 R 0.02	*p* = 0.66 R 0.07	*p* = 0.80 R −0.04	*p* = 0.80 R −0.04
** *TRA2B* ** ** *Numerical* **	***p* = 0.01 (−)** R −0.43	*p* = 0.24 R −0.29	***p* = 0.03 (+)** R 0.40	***p* < 0.01 (+)** R 0.46	*p* = 0.49 R −0.11	*p* = 0.62 R 0.08	*p* = 0.25 R 0.19	*p* = 0.25 R 0.19
** *TIA1* ** ** *Numerical* **	***p* = 0.06 (−)** R −0.33	***p* = 0.04 (−)** R −0.44	***p* = 0.04 (+)** R 0.35	***p* = 0.01 (+)** R 0.45	*p* = 0.27 R −0.19	*p* = 0.91 R 0.01	*p* = 0.68 R −0.07	*p* = 0.68 R −0.07

Abbreviations: DOI, Depth of Invasion; DOIx3 (1–5 mm, 5–10 mm. >10mm); G, grade; Invasion front [expansive (+) vs. infiltrative (−)]; LVI, lymphovascular invasion; pN, cervical metastasis (pN0/pN1/pN2a/pN2b/pN3); pNx4 (pN0/pN1/pN2/pN3); pNx2 (pN0 + pN1/pN2 + pN3), pN- (pN0) vs. pN+ (pN1, pN2, pN3); PNI, perineural invasion; pT, tumor size (pT1, pT2, pT3, pT4); pTx2 (pT1 + pT2/pT3 + pT4); PTI (mild, moderate, severe), PTIx2 (absent + mild/moderate + severe); Stage (I/II/III/IV); Stage x2 (I + II/III +/IV); Uniformity [poorly defined tumor edges (−) vs. well-defined edges (+)]; (−), negative correlation; (+), positive correlation.

**Table 4 ijms-25-06929-t004:** In vivo relationship between the expression of spliceosome components in OSCC and lymph node pathological data, Extranodal Extension (ENE), and node size. A Spearman correlation test was used to analyze the numerical expression of spliceosome components and lymph node results. (−), negative correlation.

	*Nº Lymph Nodes*	*Nº ENE+*	*Size* (mm)
** *SRSF4* ** ** *Numerical* **	*p* = 0.10 R −0.32	*p* = 0.45 R −0.15	*p* = 0.19 R −0.25
** *SRSF5* ** ** *Numerical* **	***p* = 0.03 (−)** R -0.38	***p* = 0.03 (−)** R -0.38	***p* = 0.07 (−)** R −0.32
** *SRSF9* ** ** *Numerical* **	*p* = 0.12 R −0.28	*p* = 0.77 R −0.05	*p* = 0.29 R 0.19
** *SRSF10* ** ** *Numerical* **	*p* = 0.75 R −0.06	*p* = 0.58 R 0.11	*p* = 0.91 R 0.02
** *NOVA1* ** ** *Numerical* **	*p* = 0.17 R −0.25	*p* = 0.32 R −0.18	*p* = 0.45 R −0.14
** *RBM3* ** ** *Numerical* **	*p* = 0.29 R −0.20	*p* = 0.54 R −0.12	*p* = 0.54 R −0.12
** *RBM10* ** ** *Numerical* **	*p* = 0.17 R −0.25	*p* = 0.74 R −0.06	*p* = 0.77 R −0.05
** *ESRP1* ** ** *Numerical* **	*p* = 0.55 R −0.11	*p* = 0.76 R 0.05	*p* = 0.65 R 0.08
** *ESRP2* ** ** *Numerical* **	*p* = 0.18 R −0.24	*p* = 0.50 R −0.12	*p* = 0.69 R −0.07
** *TRA2A* ** ** *Numerical* **	***p* = 0.01 (−)** R −0.48	*p* = 0.22 R −0.24	***p* = 0.04 (−)** R −0.39
** *TRA2B* ** ** *Numerical* **	***p* < 0.01 (−)** R −0.55	***p* = 0.04 (−)** R −0.38	***p* = 0.01 (−)** R −0.44
** *TIA1* ** ** *Numerical* **	***p* < 0.01 (−)** R −0.57	***p* = 0.09 (−)** R −0.33	***p* = 0.01 (−)** R −0.47

## Data Availability

The datasets generated and/or analyzed during the current study are available from the corresponding author upon reasonable request.

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
