# Peer review of "Splicing Machinery Is Impaired in Oral Squamous Cell Carcinomas and Linked to Key Pathophysiological Features"

_ijms, 2024, doi:10.3390/ijms25136929_

Round 1

Reviewer 1 Report

Comments and Suggestions for Authors

The authors looked into oral squamous cell carcinomas using cancer patients and normal tissue. They are, however, focused on the splicing pathway. They show overall changes with respect to oral cancer and show that using an inhibitor of the splicing pathway on cancer from patient tissue. Overall, it is interesting work; however, some of the concerns are below:

1.    The author could provide the details of the patients in a separate table. The Cox regression should show the Hazard ratio/odds ratio with confidence intervals.

2.    The authors find that most of the splicing factors are downregulated in the patients. Did they observe similar things in publicly available datasets? The upregulation of SRSF10 seems relatively small. It does not affect OS/RR etc. Incidentally, some of the downregulated factors are significant in Table 1. How do the authors support this? Is overall splicing downregulated in OSCC? Additionally, were these adjusted for stage, grade etc?

3.    Did the author try western blotting for tumor-normal tissue? Or siRNA in a cell line, possibly. The inhibitor seems to be specific for another member of the splicing family. Can they show specificity for the SRSF10? The tumor-normal cell comparison seems to show a difference, but that could be due to the doubling of the time for cancer vs. normal cells.

4.    The authors should explain the normalization factor more in Figure 1. Is this the same array from fluidigm that they used?

Author Response

Point-by-point responses to the Reviewers’ comments

 We sincerely thank the Editor and the Reviewers for their constructive comments, which we found very helpful towards improving the quality of our study. Accordingly, specific changes have been made in the manuscript, based on these comments, as it is described in detail below in a point-by-point description of the changes introduced, and how Reviewer’s concerns were addressed. Changes within the manuscript are indicated in red. We honestly trust that our new results and responses will help to strengthen the support of the Reviewers and hope that this revised version of our manuscript fits within the high-quality scientific standards of International Journal of Molecular Sciences and may therefore become acceptable for publication in the journal.

Responses to Reviewer 1: We thank the Reviewer for the positive comments, and for the helpful criticisms and corrections provided. The following are responses to the specific comments raised:

Comment 1: The author could provide the details of the patients in a separate table. The Cox regression should show the Hazard ratio/odds ratio with confidence intervals.

 Author´s response:  We thank the Reviewer for this pertinent comment. Following the reviewer´s request, we have included details of the patients at the beginning of the Materials and Methods section, as follows (lines 370-375): “This study includes the analysis of 37 patients diagnosed with OSCC, 19 men (52%) and 18 women (48%), with a mean age of 64±2-years-old (range 26-86 years). SCC originated from the tongue in 20 out of 37 patients (54%) and from the floor of the mouth in 6 patients (16%). In the rest, it was found in the alveolar ridge or hard palate in 5 patients (14%), in the buccal mucosa in 3 patients (8%), in the retromolar trigone in 2 patients (5%), and in 1 patient (3%) the origin was the lower lip”.

In term of the Cox regression, we agree with the reviewer that a Proportional Hazard Model would be an ideal way to expose the survival results; however, the small sample of the study precluded us to perform this specific multivariable analysis. Therefore, we plan to continue increasing the number of samples for future investigations and continue following these patients for a proper analysis of the impact of the splicing machinery components on patient´s survival. However, these planned studies will necessarily require some time (and associated extra funding) and are meant to be the subject of independent manuscripts in the future analyzing the specific role of key splicing machinery components in OSCC. In this sense, although this study cannot assure that splicing factors have a true impact on the overall survival, we want to emphasize the novel results included in the manuscript demonstrating that the dysregulated levels of the expression of some splicing machinery components in OSCC (i.e. SST2) are significantly associated with prognosis factors and histopathological tumor features, and that the treatment with a splicing machinery inhibitor can significantly decreased proliferation rate of OSCC cells. Accordingly, following the review´s comment, we have included this as a limitation of the study (last paragraph of the discussion; line 345), and we would respectfully request to the Reviewer that this additional and challenging data not to be considered as a requisite for the potential acceptance of our present revised manuscript.

Comment 2a: The authors find that most of the splicing factors are downregulated in the patients. Did they observe similar things in publicly available datasets?

Author´s response: We appreciate this insightful observation. This information is included in the discussion section (lines 265- 274) and in Table 4, as follows: “the expression levels of specific splicing factors in OSCC samples vary in the literature (Table 4). Some splicing factors have been described as upregulated, downregulated, or even oppositely altered 3,10,32,50. Our study found an overall downregulation of most of the splicing factors in OSCC samples, inviting us to explore these molecules further as potential diagnostic and prognostic biomarkers (see below for further discussion). These results were consistent with prior studies that also found down-regulation of NOVA136, RBM335, ESRP1, and ESRP234 in OSCC, but, to the best of our knowledge, this is the first study to describe the downregulation of SRSF4, SRSF9, RBM10, TRA2A, TRA2B, and TIA1 in OSCC. On the other hand, we found SRSF10 expression to be upregulated in our cohort of OSCC samples, which was also consistent with a previous study reported with head and neck samples30.

In this sense, we noticed that we erroneously mentioned Table 1 in the introduction that indeed corresponds to table 4. We have deleted that reference to Table 1 (line 52) in the introduction and indicated also in the introduction that the table 4 mentioned in the discussion includes the information related to previous findings about this topic in OSCC (lines 55-56).

Comment 2b: The upregulation of SRSF10 seems relatively small. It does not affect OS/RR etc. Incidentally, some of the downregulated factors are significant in Table 1. How do the authors support this? Is overall splicing downregulated in OSCC? Additionally, were these adjusted for stage, grade, etc?

 Author´s response: We thank the Reviewer for this comment. Yes, as previously observed in other tumoral pathologies [for instance, in pituitary tumors: Vazquez-Borrego et al., 2019. Cancers 11(10):1439], we found an overall downregulation in the expression of the splicing machinery components in OSCC compared to healthy-control tissues. As indicated in the introduction and discussion of our manuscript, this finding could be pathophysiological relevant not only in OSCC but also in different tumors since this alteration can significantly affect the expression pattern of many essential genes and the appearance of oncogenic splicing variants that have been reported to play oncogenic roles in many cancer types.

In the case of SRSF10, our data indicated that SRSF10 was the only splicing factor found to be upregulated in OSCC tissues although this difference (visually inapparent as can be observed in Figure-1B) was not as significant (*) as other splicing factors that were clearly downregulated in OSCC (** or ***) (Figure 1). This upregulation in the expression levels of SRSF10 is consistent with a previous study in head and neck cancer (see Table 4 in the discussion section– Yadav et al). Therefore, based on the reviewer´s comment, we have included a sentence in the discussion section indicating that SRSF10 upregulation did not show any correlation with OSCC survival or histopathological risk factors in the context of the previous data (lines 272-278), as follows: “On the other hand, we found SRSF10 expression to be upregulated in our cohort of OSCC samples, which was also consistent with a previous study reported with head and neck samples30. However, in our study, SRSF10 upregulation (whose change was visually inapparent as can be observed in Figure-1B) did not show any correlation with OSCC survival or histopathological risk factors, while SRSF10 was reported to play a crucial role in head and neck tumorigenesis in the previous study30. This difference may be due to the intrinsic phenotypic differences between samples, as well as protocol and race differences.”.

Comment 3a: Did author try western blotting for tumor-normal tissue? Or siRNA in a cell line, possibly?

Author´s response: We appreciate this insightful suggestion, and we have to agree with the Reviewer that this would be truly interesting. Unfortunately, due to the difficulty in obtaining this type of high-value samples and the limited of the tissue material obtained, it was not possible to obtain enough quality protein to perform western blotting. Indeed, we had to prioritize the experiments that could be performed with this limited tissue and decided to carry out expression levels using microfluidic qPCR technology (that allows to simultaneously measure and compared the expression levels of multiple genes in OSCC samples and normal healthy-adjacent tissues) and cell cultures experiments in response to the inhibitor of the splicing machinery. Accordingly, following the review´s comment, we have included this as a limitation of the study (last paragraph of the discussion; line 347), and therefore, we would respectfully request to the Reviewer that this additional and challenging data not to be considered as a requisite for the potential acceptance of our present revised manuscript.

 Comment 3b: The inhibitor seems to be specific for another member of the splincing family. Can they show specificity for the SRSF10? The tumor-normal cell comparison seems to show a difference, but that could be due to the doubling of the time cancer vs normal cells.

Author´s response: We thank the Reviewer for bringing up this interesting issue. Pladienolide-B, as previously reported, specifically targets the SF3B protein (subunit 1). Although it would have been a very interesting point, due to the lack/limitation of the tissue material obtained, it was not possible to analyse the expression profile of these cells after treatment with the inhibitor, Pladienolide-B. Based in our results, the difference observed in the proliferation analysis between normal and tumor cells seems to be exclusively triggered by the inhibitor [as we have previously reported in other cancer types such as brain tumors: Fuentes-Fayos et al., S2022. F3B1 inhibition disrupts malignancy and prolongs survival in glioblastoma patients through BCL2L1 splicing and mTOR/ss-catenin pathways imbalances. J Exp Clin Cancer Res 41(1):39]. The cell viability analysis was normalized comparing each treated cells (tumor/normal) against the same untreated cell type. By this way, the doubling of the time cancer vs. normal cells was not affected in the result showed in the present study.

Comment 4: The authors should explain the normalization factor more in Figure 1. Is this the same array from fluidigm that they used?

Author´s response: We thank the Reviewer for this important observation. Following the review´s suggestion, we have clarified this point in the revised manuscript (section 4.3 of the Materials and Methods: lines 404-405 and 410-417).

Reviewer 2 Report

Comments and Suggestions for Authors

The authors presented first their logical reasoning to attempt a study on impaired splicing machinery in oral cancer. In general splicing dysregulation was found in many cancers. However, it is perfectly known that given information is not directly transmitted to another one. Then appears another question concerning function of particular components of splicing machinery. Taking into account epidemiological data the authors decided to gain more information about splicing dysregulation in oral squamous cell carcinoma. I fully accept a background of the attempted studies.

An experiments were limited to the application of Pladienolide-B as a compound involved in splicing machinery  regulation.  Only of few studied components of splicing machinery were associated  with significant  changes of expression. The authors suggest further studies of findings into clinical application.
The only critical remars is connected with a relatively small groups of materiaÅ‚ donors. Then the group was divided further according to demographic and clinical data it went into small subgroups not necessarily providing  significant results. However, on the other hand a spectrum of differences is becoming visible.

Minor remark. Reference [30] seems to not have sufficient bibliographic information.

Author Response

Point-by-point responses to the Reviewers’ comments

 We sincerely thank the Editor and the Reviewers for their constructive comments, which we found very helpful towards improving the quality of our study. Accordingly, specific changes have been made in the manuscript, based on these comments, as it is described in detail below in a point-by-point description of the changes introduced, and how Reviewer’s concerns were addressed. Changes within the manuscript are indicated in red. We honestly trust that our new results and responses will help to strengthen the support of the Reviewers and hope that this revised version of our manuscript fits within the high-quality scientific standards of International Journal of Molecular Sciences and may therefore become acceptable for publication in the journal.

Reviewer’s General Comment: The authors presented first their logical reasoning to attempt a study on impaired splicing machinery in oral cancer. In general splicing dysregulation was found in many cancers. However, it is perfectly known that given information is not directly transmitted to another one. Then appears another question concerning function of particular components of splicing machinery. Taking into account epidemiological data the authors decided to gain more information about splicing dysregulation in oral squamous cell carcinoma. I fully accept a background of the attempted studies. An experiments were limited to the application of Pladienolide-B as a compound involved in splicing machinery regulation.  Only of few studied components of splicing machinery were associated with significant changes of expression. The authors suggest further studies of findings into clinical application. The only critical remark is connected with a relatively small groups of materiaÅ‚ donors. Then the group was divided further according to demographic and clinical data it went into small subgroups not necessarily providing significant results. However, on the other hand a spectrum of differences is becoming visible.

Author’s response: We thank the Reviewer for the laudatory comments of our work, and for the helpful comments provided. We agree that a limitation of the study is the limited number of cases analyzed that we would like to continue increasing for future investigations and following these patients for a proper analysis of the impact of spliceosome components on patient´s survival and other relevant clinical/pathological characteristic. This limitation has been included in the last paragraph of the discussion section (lines 342-345).

Comment 1: Minor remark. Reference [30] seems to not have sufficient bibliographic information.

Author´s response: We thank the Reviewer for pointing this out. We have included the full information in reference 30 (lines 553-554).

Reviewer 3 Report

Comments and Suggestions for Authors

The manuscript for San Juan-San Juan A. is well-written and interesting. The authors elucidate the relationship between splicing machinery activity and OSCC, primarily emphasizing the impact of poor OSCC. The authors showed intriguing results. That could provide essential but important information for future research, as well as new elements for planning therapies against those carcinomas. The discussion adequately explains the study's significance and its limitations. In future studies, I believe that the evaluation of SM will provide an important link between oral cancer and prognosis, as well as a treatment. 

Author Response

Point-by-point responses to the Reviewers’ comments

 We sincerely thank the Editor and the Reviewers for their constructive comments, which we found very helpful towards improving the quality of our study. Accordingly, specific changes have been made in the manuscript, based on these comments, as it is described in detail below in a point-by-point description of the changes introduced, and how Reviewer’s concerns were addressed. Changes within the manuscript are indicated in red. We honestly trust that our new results and responses will help to strengthen the support of the Reviewers and hope that this revised version of our manuscript fits within the high-quality scientific standards of International Journal of Molecular Sciences and may therefore become acceptable for publication in the journal.

Reviewer’s General Comment: The manuscript for San Juan-San Juan A. is well-written and interesting. The authors elucidate the relationship between splicing machinery activity and OSCC, primarily emphasizing the impact of poor OSCC. The authors showed intriguing results. That could provide essential but important information for future research, as well as new elements for planning therapies against those carcinomas. The discussion adequately explains the study's significance and its limitations. In future studies, I believe that the evaluation of SM will provide an important link between oral cancer and prognosis, as well as a treatment.

Author’s response: We thank the Reviewer for the laudatory comments of our work.